# Spatially Resolved Gene Expression Prediction from H&E Histology Images via Bi-modal Contrastive Learning

**Ronald Xie**[1,2,3,4]    **Kuan Pang**[1,2,4]    **Sai W. Chung**[1,5]    **Catia T. Perciani**[1,5]
**Sonya A. MacParland**[1,5]    **Bo Wang**[1,2,3*]    **Gary D. Bader**[1,3,4,6*]

[1]University of Toronto, [2]Vector Institute, [3]University Health Network,
[4]The Donnelly Centre, [5]Toronto General Hospital Research Institute,
[6]Canadian Institute for Advanced Research (CIFAR)
{ronald.xie, kuan.pang, sai.chung,
catia.perciani, gary.bader}@mail.utoronto.ca,
Sonya.MacParland@uhnresearch.ca, bowang@vectorinstitute.ai

## Abstract

Histology imaging is an important tool in medical diagnosis and research, enabling the examination of tissue structure and composition at the microscopic level. Understanding the underlying molecular mechanisms of tissue architecture is critical in uncovering disease mechanisms and developing effective treatments. Gene expression profiling provides insight into the molecular processes underlying tissue architecture, but the process can be time-consuming and expensive. We present BLEEP (Bi-modaL Embedding for Expression Prediction), a bi-modal embedding framework capable of generating spatially resolved gene expression profiles of whole-slide Hematoxylin and eosin (H&E) stained histology images. BLEEP uses contrastive learning to construct a low-dimensional joint embedding space from a reference dataset using paired image and expression profiles at micrometer resolution. With this approach, the gene expression of any query image patch can be imputed using the expression profiles from the reference dataset. We demonstrate BLEEP's effectiveness in gene expression prediction by benchmarking its performance on a human liver tissue dataset captured using the 10x Visium platform, where it achieves significant improvements over existing methods. Our results demonstrate the potential of BLEEP to provide insights into the molecular mechanisms underlying tissue architecture, with important implications in diagnosis and research of various diseases. The proposed approach can significantly reduce the time and cost associated with gene expression profiling, opening up new avenues for high-throughput analysis of histology images for both research and clinical applications.
Code available at `https://github.com/bowang-lab/BLEEP`

## 1   Introduction

Histology imaging of whole-slide Hematoxylin and eosin (H&E) stained tissues has long been used in academic and clinical settings for research and diagnosis. It provides useful information pertaining to tissue architecture and composition at the microscopic level, which are critical for understanding disease mechanisms and developing effective treatments. Gene expression profiling is a powerful tool that offers deeper insights into the molecular processes underlying tissue architecture. However,

---

[*]Co-senior author

37th Conference on Neural Information Processing Systems (NeurIPS 2023).

bulk RNA sequencing does not capture heterogeneity within one sample whereas single-cell RNA sequencing (scRNA-seq) or single-nucleus RNA sequencing (snRNA-seq) captures heterogeneity without spatial context.

In recent years, various spatial transcriptomics methods such as Visium[22], MERFISH [4], seqFISH+[8], STARmap[23], smFISH[6], and Targeted ExSeq[1] have emerged as a promising direction to bridge the gap between histology imaging and gene expression profiling. However, these methods are often low throughput or low content. They also tend to be time-consuming, expensive and require specialized equipment and extensive domain expertise to optimize.

With the availability of these bi-modal datasets, a unique opportunity arises to explore the possibility of predicting spatially resolved expression profiles of whole tissues solely from their histology image. ST-Net and HisToGene are two methods developed for this purpose [14, 9] but so far have achieved limited success. Current progress has been limited due to three significant challenges. Firstly, the problem is ill-posed. While the histology images share some information with their paired spatial transcriptomics, it is likely that the image features can not be used to predict the expression of all genes and vice versa. However, expression of marker genes for cell types or subtypes (MG), highly expressed genes (HEG), and highly variable genes (HVG) should be prioritized as they tend to be the most biologically relevant candidates for disease diagnosis and drug development. Secondly, the problem is dimensionally cursed. It is estimated that typical mammalian cells express around 5,000 to 15,000 genes. Existing solutions often only predict the expression of a limited panel of genes (~200) and often fail to preserve the variance and heterogeneity in the original dataset. This obfuscates the biological signals intrinsic in the original data, rendering the predictions ineffective for practical use. Thirdly, due to the developing landscape of spatial transcriptomic methods, existing datasets are prone to experimental artifacts both within one sample and across different samples, which complicates the training of expression prediction models. Furthermore, the measured gene expressions often show poor agreement with their protein profiles, which could be reflected during H&E staining.

We present BLEEP (Bi-modaL Embedding for Expression Prediction), a novel bi-modal embedding framework designed to address the aforementioned challenges associated with predicting gene expression from histology images. BLEEP uses contrastive learning, which effectively aligns paired image and expression representations from a reference dataset in a low-dimensional joint embedding space. Using this joint embedding space, BLEEP accurately imputes the gene expression of any query image patch by leveraging the expression profiles from the reference dataset.

We demonstrate the effectiveness of BLEEP by benchmarking its performance using a challenging human liver tissue dataset captured via the 10x Visium platform, where it significantly outperforms existing methods such as HisToGene and ST-Net in terms of the average correlation to original expressions across marker genes (MG), highly expressed genes (HEG) and highly variable genes (HVG). BLEEP also preserves heterogeneity in the predicted expression profiles and recaptures and denoises the gene-gene correlations present in the original dataset.

The proposed approach alleviates the ill posed nature of the expression prediction problem by implicitly encouraging shared features to be learned between image and expression modalities via a contrastive objective which could prevent modality specific information from disorienting the joint embedding space. The novel query-reference imputation process from the learned joint embedding serve to mitigate the curse of dimensionality of the expression prediction problem as the gene expression profiles of query image patches are no longer predicted independently, but rather calculated from $k$ closest reference expression profiles in the joint space. Lastly, we demonstrate that the resulting expression prediction is resilient to experimental artifacts both within one sample and across different samples, in addition to outperforming existing solutions on the aforementioned metrics.

To our knowledge, this is the first bi-modal embedding-based framework proposed for the task of expression prediction from histology images. The proposed approach has the potential to significantly reduce the time and cost associated with gene expression profiling, opening up new avenues for high-throughput analysis of histology images for both research and clinical applications.

## 2 Related Works

### 2.1 Existing histology expression prediction approaches

Several existing approaches have shown promising results in predicting expression from histology images including HE2RNA[20], ST-Net[9], HisToGene[14], hist2rna[13], Hist2ST[26] and others[7, 24].

ST-Net and HisToGene are two of the most popular methods for predicting spatially resolved expression from H&E images. Both of these approaches frame the task of expression prediction as regression tasks trained in a feed-forward fashion. ST-Net uses a resnet50 image encoder followed by a fully connected layer where as HisToGene leverages a vision transformer backbone and an increased field of view.

Methods that predict tissue-level expression generally achieve good correlation but lack the ability to generate spatially resolved expression profiles (HE2RNA). Existing methods that are capable of generating spatially resolved expression predictions were either not quantitatively evaluated (hist2RNA), limited in terms of the predicted panel (ST-Net, Hist2ST, HisToGene), or prone to overfitting [24].

Both HisToGene and Hist2ST utilize spot-spatial relations to improve performance. However, our work challenges the necessity of this information, particularly in tissues with distinct and repetitive spatial patterns like human liver tissue. The implicit assumption that spatially adjacent regions should have similar representations compared to spatially distant regions may not be beneficial for performance in such cases. Hard coding position information could also lead to overfitting in data-scarce scenarios.

### 2.2 Contrastive representation learning

Contrastive learning plays a pivotal role in advancing the capabilities of deep learning models, particularly in recent visual language models [5, 18, 17, 15]. One notable application that has emerged from contrastive learning is the Contrastive Language-Image Pretraining (CLIP) framework [15]. CLIP bridges the gap between language and vision domains by learning joint representations of paired images and textual descriptions, enabling cross-modal understanding and reasoning.

BLEEP draws inspiration from CLIP with modifications to learn a similar joint embedding between spot expression profiles captured by the 10x Visium platform and their spatially paired image patch spanning roughly $55\mu$m. However, unlike CLIP's usage setting, BLEEP directly interpolates in the joint embedding space to produce expression predictions, which is unfeasible for image and text domains where a domain-specific decoder is required to produce the final prediction.

### 2.3 Query-reference imputation

The query-reference imputation of BLEEP is partly inspired by SeuratV3's [19] integration process, where the expression profiles are calculated from a linear combination of the closest anchors in the reference dataset given a query. However, Seurat requires a shared expression panel across modalities to integrate, whereas BLEEP is able to make spatially resolved expression predictions based on the morphological features present in the histology image alone.

## 3 Methods

### 3.1 Data and preprocessing

The dataset [2, 3] used to train and benchmark BLEEP, HisToGene, and ST-Net consists of four consecutive 16 micrometer thick slices of human liver tissue from neurologically deceased donor livers suitable for transplantation that were OCT embedded, frozen, sliced with a cryostat and imaged using the 10x Genomics Visium platform [3]. Samples were collected with institutional ethics approval from the University Health Network (REB# 14-7425-AE). After quality control, the slices contain 2378, 2349, 2277, and 2265 spots respectively. A $224{\times}224$ image patch centered around each spot roughly $55\mu$m each side is extracted from the whole slide H&E image and paired with the

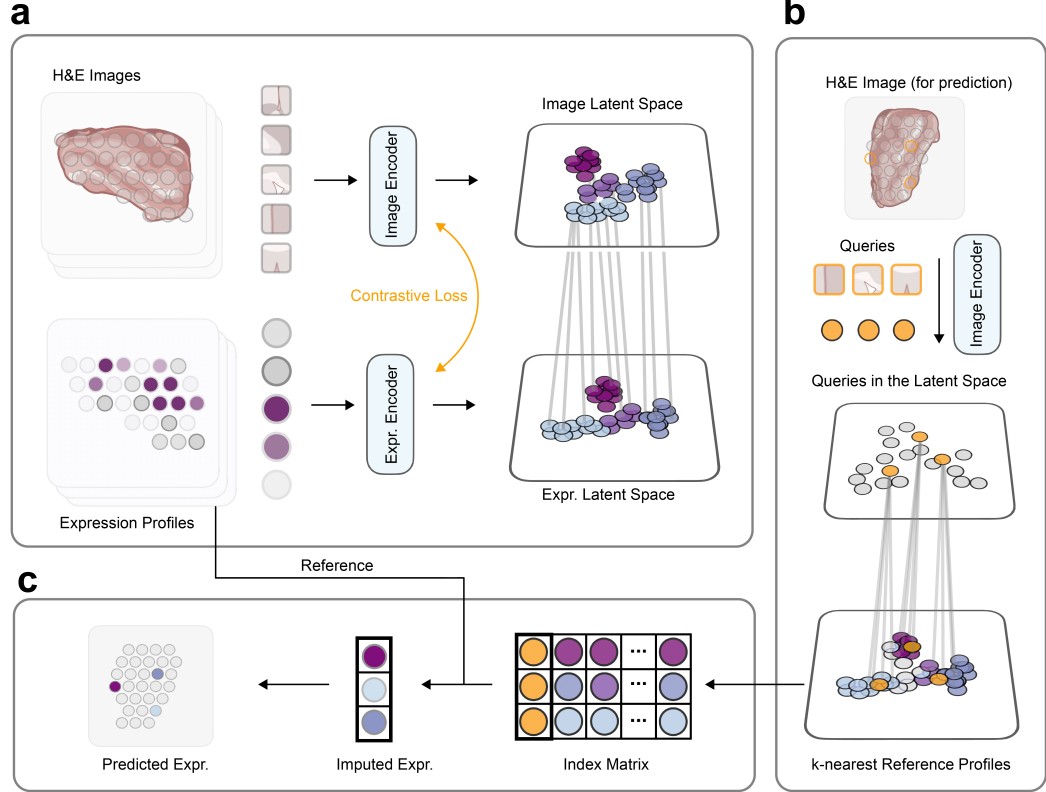

Figure 1: BLEEP achieves gene expression prediction from H&E image through (a) BLEEP learns a bimodal embedding from expression profiles and H&E image patches, (b) images patch queries are projected into the joint embedding space to index the $k$ nearest reference expression profiles, and (c) the indexed reference expression profiles are linearly combined to produce the imputed gene expressions for queries.

corresponding gene expression profile of each spot. The data is publicly available for download at `https://www.ncbi.nlm.nih.gov/geo/query/acc.cgi?acc=GSE240429`.

Each spot is normalized to the total count and log normalized before highly variable genes are computed using the Scanpy[25] package. The union of the top 1000 most highly variable genes from each of the 4 slices was used for training and prediction, amounting to 3467 genes in total. Finally, the expression data of these four samples are batch corrected using Harmony [11] before one of the slices (slice #3) is randomly selected to be held out for testing.

## 3.2   Learning bimodal embedding for expression prediction

As illustrated in Figure 1a, the first step towards learning a bimodal embedding for expression prediction is to extract features from the two modalities respectively using the image encoder and the expression encoder. Given a batch of $B$ paired image patches ($V \in \mathbb{N}^{B \times L \times L}$) and normalized expression profiles ($X \in \mathbb{N}^{B \times C}$), where $L$ is the image patch size and $C$ is the gene set size, we use an image encoder $f_{img}$ and an expression encoder $f_{expr}$ to project the inputs into $h$-dim image embeddings and expression embeddings through $H_v = f_{img}(H) \in \mathbb{N}^{B \times h}$ and $H_x = f_{expr}(X)\mathbb{N}^{B \times h}$. We use contrastive learning to further align the latent space for $H_v$ and $H_x$.

Our contrastive learning approach is inspired by CLIP [16, 21], but the adjusted loss function better smooths out the loss landscape for our specific use case. Unlike natural images, multiple spots with similar expression profiles or image morphology are often expected to be sampled in the same batch. The modification will prevent the model from pulling apart spots with similar expression profiles

---
**Algorithm 1** Bimodal Embedding for Expression Prediction

---
**Input**: Paired image patches ($V \in \mathbb{N}^{B \times L \times L}$), normalized expression profiles ($X \in \mathbb{N}^{B \times C}$), image encoder $f_{img}$, expression encoder $f_{expr}$, temperature $\tau$
**Output**: Joint embedding space ($H_v, H_x$)

---
 1: **function** BIMODALEMBEDDING($V, X, f_{img}, f_{expr}, \tau$)
 2:    $H_v, H_x \leftarrow f_{img}(V), f_{expr}(X)$           ▷ Image and Expression embeddings
 3:    $sim(H_v, H_x) \leftarrow H_x \cdot H_v^T$              ▷ Paired similarity
 4:    $sim(H_v, H_v) \leftarrow H_v \cdot H_v^T$            ▷ Internal similarities (image)
 5:    $sim(H_x, H_x) \leftarrow H_x \cdot H_x^T$         ▷ Internal similarities (expression)
 6:    $target \leftarrow \mathrm{softmax}((sim(H_x, H_x) + sim(H_v, H_v))/2 \cdot \tau)$     ▷ Similarity-adjusted target
 7:    $\mathcal{L} \leftarrow \mathrm{mean}(\mathrm{ce}(sim(H_v, H_x), target) + \mathrm{ce}(sim(H_v, H_x)^T, target^T))$ ▷ Cross entropy loss
 8:    **return** $H_v, H_x, \mathcal{L}$
 9: **function** QUERYREFERENCEIMPUTATION(Image, $f_{img}, H_v, X, k$)
10:    $V' \leftarrow \mathrm{split}(Image)$                   ▷ Split into image patches
11:    $Q'_v \leftarrow f_{img}(V')$            ▷ Project patches(queries) into embedding space
12:    $distances \leftarrow \mathrm{dist}(q', H_v, l_2)$ for $q' \in Q'_v$     ▷ Compute l2 distance for each query
13:    $indices \leftarrow \mathrm{topk}(distances)$    ▷ Get indices of top K matched expression from reference
14:    reference_profiles $\leftarrow X[indices]$
15:    **return** weighted_avg(reference_profiles)            ▷ Return prediction

---

within the same batch, thereby increasing the coherence of the resulting joint embedding. We use an existing implementation of this loss variant[21]. In detail, we first generate the paired similarity through $sim(H_v, H_x) = H_x H_v^T$. To take into account the pairs with similar morphological features or expression landscapes, we begin with calculating the internal similarities, $sim(H_v, H_v) = H_v H_v^T$ and $sim(H_x, H_x) = H_x H_x^T$, then a similarity adjusted target matrix is denoted as:

$$target = (softmax(sim(H_x, H_x) + sim(H_v, H_v))/2 \cdot \tau)$$

where $\tau$ is a temperature hyperparameter. Cross entropy(ce) loss is applied to align the image features and expression features to produce the final loss $\mathcal{L}$:

$$\mathcal{L} = mean(ce(sim(H_v, H_x), target) + ce(sim(H_v, H_x)^T, target^T))$$

For BLEEP, we use the pretrained ResNet50[10] as the image encoder and a fully connected network (FCN) with an output dimension of 256 as the expression encoder, which doubles as a projection head. The image features from the image encoder are passed through a separate projection head to bring the two modalities to the same dimension before applying the contrastive loss similar to CLIP[15], where the model learns to pull the paired representations together while pushing other representations apart. We find that the ResNet50 image encoder with fewer trainable parameters obtained more favorable results compared to various pretrained vision-transformer (ViT) encoders (Supplementary Table 1). Larger models in conjunction with a relatively small training dataset may encourage information to be memorized in the weights of the network rather than being encoded in the projections, therefore rendering the learned joint embedding ineffective for downstream imputation for our use case.

BLEEP is trained using 4 NVIDIA V100 GPUs with the AdamW optimizer[12], a batch size of 512 and a learning rate of 0.001 for 150 epochs.

### 3.3 Query-Reference imputation

As illustrated in Figure 1b, the process starts by first splitting the H&E image into $N$ small image patches to be encoded by the trained image encoder. Once the image patches are represented in the joint embedding space, the $k$ nearest expression profiles from the reference are selected based on their proximity (by Euclidean distance) in the joint embedding space to each patch. Finally, the expression profiles of the query patches are imputed as a linear combination of the selected expression profiles in the reference 1c. Refer to supplementary materials for implementation details.

Table 1: Average correlation of predicted expression for 8 marker genes derived from Andrews et al. [2] (MG), top 50 most highly expressed genes (HEG) and top 50 most highly variable genes (HVG) compared to ground truth expressions on held out dataset.

| Method | MG | HEG | HVG |
|--------|-----|-----|-----|
| HisToGene | $0.097\pm0.015$ | $0.072\pm0.018$ | $0.071\pm0.011$ |
| ST-Net | $0.099\pm0.020$ | $0.126\pm0.005$ | $0.091\pm0.007$ |
| BLEEP | **0.217**$\pm0.002$ | **0.175**$\pm0.016$ | **0.173**$\pm0.011$ |

Table 2: Predicted gene expression values with top 5 correlations with original profile for each method from one representative replicate.

| HisToGene | | ST-Net | | BLEEP | |
|-----------|---|--------|---|-------|---|
| Gene Name | r | Gene Name | r | Gene Name | r |
| CYP3A4 | 0.549 | CYP3A4 | 0.549 | CYP3A4 | 0.741 |
| CYP1A2 | 0.542 | CYP1A2 | 0.532 | CYP1A2 | 0.681 |
| GLUL | 0.488 | CYP2E1 | 0.530 | CYP2E1 | 0.675 |
| CYP2E1 | 0.330 | GLUL | 0.463 | GLUL | 0.656 |
| FABP1 | 0.328 | SLCO1B3 | 0.375 | FABP1 | 0.503 |

## 4  Experiments

### 4.1  BLEEP predicts spatially resolved gene expression profiles that correlate well with original expression

Table 1 shows the performance of HisToGene, ST-Net and BLEEP for predicting a marker gene set (MG) derived from literature by Andrews et al. [2], the top 50 most highly expressed genes (HEG) and the top 50 most highly variable genes (HVG). The BLEEP predicted expression profiles show the highest correlation with ground truth across all three gene sets, achieving an increase of 120%, 39% and 90% in $r$ value across the three gene sets, respectively, compared to the second scoring method.

Furthermore, Table 2 shows the top 5 predicted genes from each method. We observed that the most well-predicted genes are relatively consistent across methods. These genes are known to be spatially zonated, with CYP3A4, CYP1A2, CYP2E1, GLUL being pericentrally zonated and FABP1 being periportally zonated. We also observe that BLEEP consistently achieved higher correlation values for these genes compared to HisToGene and ST-Net, further demonstrating the effectiveness of BLEEP in expression prediction.

Despite the good prediction of select genes in Table 2, we note that the overall absolute correlation in Table 1 remains low, highlighting the difficulty of the prediction task for most genes. The low scores could be attributed to several causes, including the expression of certain genes being poorly correlated with morphological features; the poor detection of certain genes by the Visium platform causing their expression to be less predictable; and experimental artifacts that could introduce non-biological variance to the data independent of the image.

### 4.2  BLEEP retains biological heterogeneity

Figure 2 highlights the key advantage of BLEEP compared to supervised regression-based approaches such as HisToGene and ST-Net. We observe that while BLEEP only narrowly outperforms HisToGene and ST-Net in correctly predicting the mean expression of genes within one sample, both HisToGene and ST-Net fail to recapitulate the variance of the genes being predicted.

While the variance of BLEEP expression predictions is in general underestimated, they still maintain sufficient biological heterogeneity particularly when the predicted expressions are plotted in fixed scale with the original expression 3. Additional figures depicting the spatial expression of other genes are available in (Supplementary Figure 1). Both HisToGene and ST-Net fail to resemble the original expression, likely due to the curse of dimensionality of the prediction task, resulting in the model

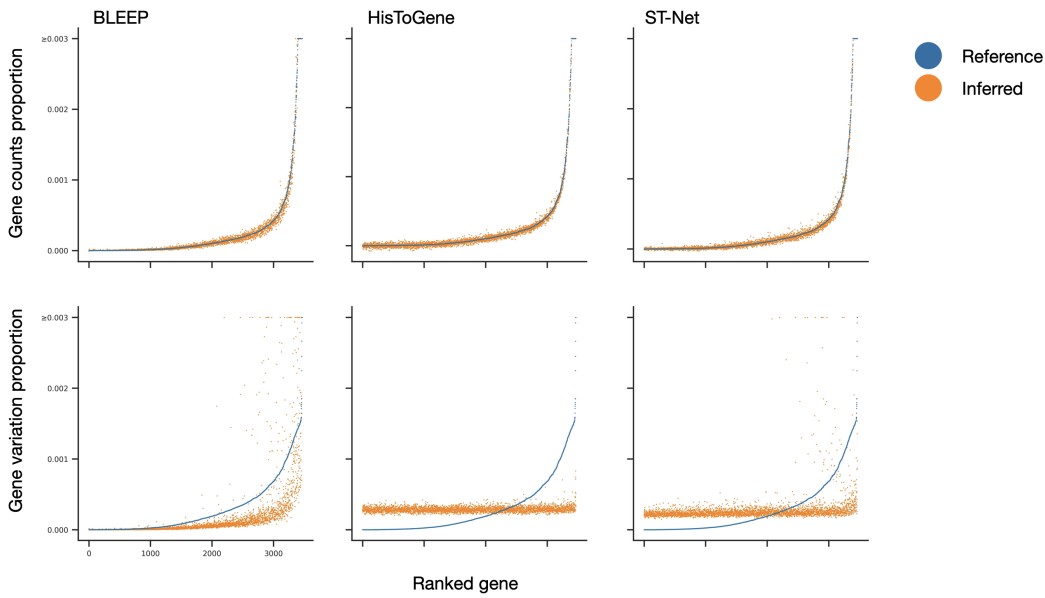

Figure 2: Predicted expression profiles compare with reference expression profiles normalized by gene count means (Upper) or gene count variance (Lower).

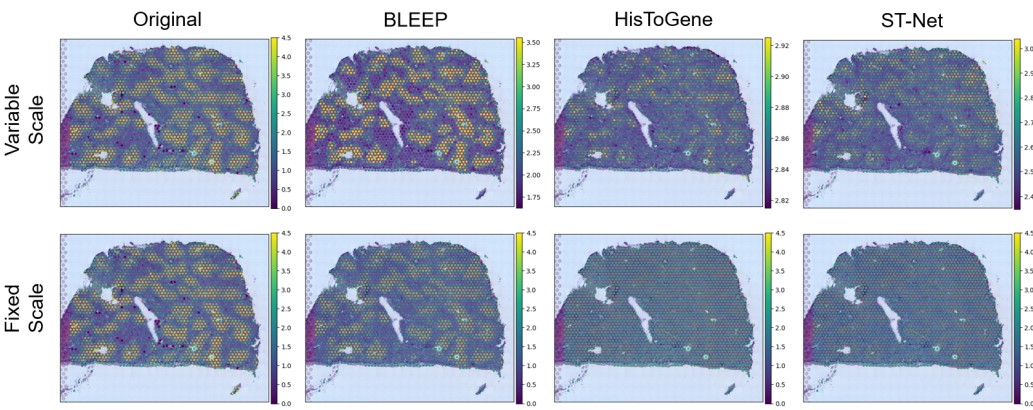

Figure 3: Original and predicted spatially resolved expression levels for CYP3A4 overlaying the H&E image, visualized with variable (Top) and fixed (Bottom) color scale.

learning mean expressions as a shortcut, with each gene having roughly the same variance regardless of their mean expression, which is undesirable.

Figure 4 demonstrates the effectiveness of BLEEP in preserving gene-gene correlations (GGCs) as further testament to its ability to preserve relevant biological heterogeneity.

### 4.3 BLEEP inferred expression is robust to experimental artifacts and batch effects

Supplementary Table 2 illustrates the clustering statistics of the unsupervised clusters resulting from the predicted expression profiles. When the resulting unsupervised clusters are projected onto the histology image in Figure 5 we make two observations. Firstly, perhaps unsurprisingly, all three methods are robust to experimental artifacts within one slice (The red regions) as the histology image surrounding the region was not affected. The low-quality regions are characterized by a lower-than-

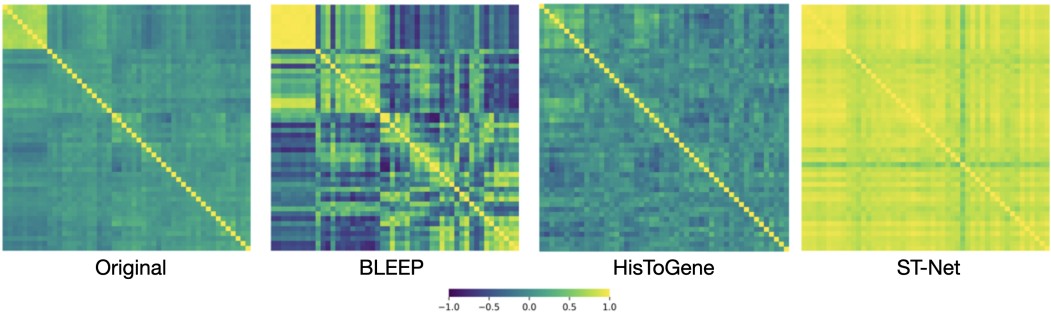

Figure 4: Gene-gene correlation heatmap calculated using the predicted expressions for each method.

normal number of counts detected and a higher-than-normal fraction of mitochondrial genes. Their expression profiles can be rescued by any method that predicts expression from the image. Secondly, all three methods are capable of capturing the zonation patterns across the tissue slice to a reasonable extent. While HisToGene has the best clustering agreement out of the three methods according to Table (Supplementary Table 2) with BLEEP and ST-Net following close behind, for this particular dataset, the clustering metrics such as NMI and ARI are likely not accurate measures of prediction quality because the hepatocytes already closely resemble each other and lie along a continuous gradient. Hence, the resulting clusters will be very sensitive to the choice of the clustering method or the hyperparameters of the method.

However, unlike HisToGene and ST-Net, BLEEP is the least prone to potentially introduce batch effects between samples during the prediction process owing to the imputation strategy. We see from Supplementary Figure 1 that both ST-Net and especially HisToGene predictions are a bit out of distribution when their predicted expression profiles are plotted alongside that of all the reference expression profiles.

## 4.4 BLEEP ablation experiments

Here we present the ablation experiments conducted in Table 3. From these experiments we make a few important observations:

The choice of K during the query-reference imputation process influences the prediction quality quite negatively when a low value is selected (K = 10). Values of K above our default value could provide some small improvements to correlation of the resulting predicted expression values for the HVG and HEG gene sets, but the differences were not pronounced. This is inline with what one might expect from taking the pseudo-bulk of the top K most likely expression profiles given any query image. However, the MG gene set did not show much improvement with increasing K. Furthermore, doing so may carry a trade off of further systematically deviating from the original variance of the dataset due to the increased averaging effect. With this in mind, we feel our default value of K = 50 remains adequate.

We also observe that the most similar match between query and reference is usually not the best prediction (as seen from the 3rd row of the ablation table and indirectly the 4th row when predictions are weighted by their similarity). We suspect the gap may close to some degree as the reference grows further in size, but in general some amount of averaging is desired for query-reference imputation to remove some noise intrinsic in the Visium platform. In addition, as no reference expression profile will perfectly match that of the query image, averaging multiple profiles is required to more appropriately describe the query expression profile. Nevertheless, we also highlight the possibility of genuine biological signals being averaged out, which is an important consideration to be further investigated.

Smoothing the contrastive loss objective to take into account patch similarity showed modest increase in performance. The gain in performance may be due to the fact that relaxing the contrastive objective is more compatible with the similarity based inference strategy. The smoothing may help lessen the extent similar references are pushed apart in embedding space during training, resulting in improved

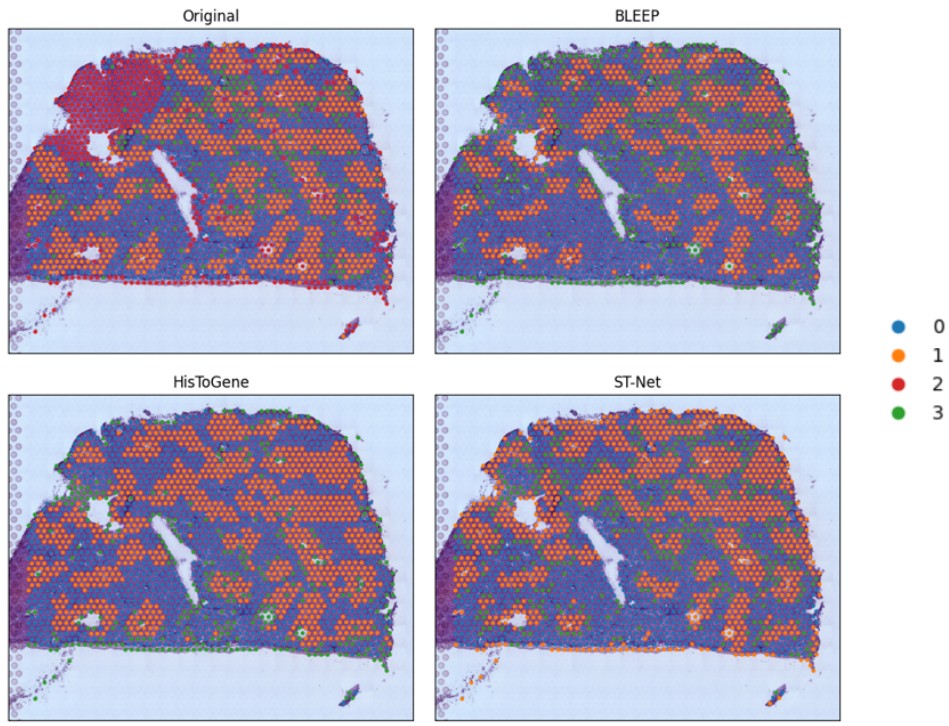

Figure 5: Leiden clusterings for original and predicted expressions overlaying the H&E image. Image-only expression predictions are invariant to low quality regions (red) during actual experiment.

Table 3: BLEEP ablation experiments. Variables tested include: smoothed objective versus the original CLIP objective, choice of top K and the method of aggregation. Pearson correlation values are reported from the marker gene set (MG), the top 50 highly variable genes (HVG) and the top 50 highly expressed genes (HEG). The correlation values show the average of 3 replicates. Accompanied uncertainty values denote the maximum difference from the mean of the 3 replicates.

| | | | Average Pearson Correlation | | |
|---|---|---|---|---|---|
| Smoothed Obj. | K | Aggregation | MG | HVG | HEG |
| Yes | 10 | average | $0.179 \pm 0.020$ | $0.146 \pm 0.008$ | $0.148 \pm 0.022$ |
| Yes | 100 | average | $0.215 \pm 0.011$ | $\mathbf{0.180 \pm 0.012}$ | $\mathbf{0.181 \pm 0.015}$ |
| Yes | – | simple | $0.079 \pm 0.032$ | $0.075 \pm 0.016$ | $0.084 \pm 0.023$ |
| Yes | 50 | weighted | $0.186 \pm 0.018$ | $0.161 \pm 0.015$ | $0.157 \pm 0.026$ |
| No | 50 | average | $0.209 \pm 0.017$ | $0.165 \pm 0.007$ | $0.170 \pm 0.005$ |
| Yes | 50 | average | $\mathbf{0.217 \pm 0.002}$ | $\mathbf{0.175 \pm 0.016}$ | $\mathbf{0.173 \pm 0.011}$ |

querying of the top K most likely expression profiles given a query image patch during the inference stage.

## 5 Discussion and Conclusion

In this study, we introduced BLEEP (Bi-modaL Embedding for Expression Prediction), a novel framework for predicting gene expression from histology images. BLEEP constructs a joint embedding space from paired image and expression features in a reference dataset and subsequently utilizes the k most similar expression profiles in the joint space to impute the expression for any given image query. To the best of our knowledge, this is the first bi-modal embedding-based framework proposed for the task of expression prediction from histology images. We also show that the query-reference

imputation process is effective and well suited for the task of expression prediction from the bi-modal joint embedding.

We demonstrated that BLEEP effectively addresses three major challenges in the H&E image to expression prediction task. Firstly, the ill-posed nature of the problem is alleviated by the proposed joint image and expression embedding space, optimized using a contrastive learning objective inspired by CLIP. This encourages the shared features between the two modalities to be preferentially encoded in the joint space. Secondly, the curse of dimensionality is tackled by the query-reference imputation process, which predicts the entire expression profile jointly through linear combination rather than predicting each individual gene separately using supervised regression tasks. Lastly, we showed that BLEEP's expression prediction is resilient to experimental artifacts, both within a single sample and across different samples.

We further observed that the correlation matrix of BLEEP's predicted expressions not only captures existing patterns in gene-gene correlations (GGCs) but also accentuates more subtle positive and negative GGCs. This could be attributed to the ability of BLEEP's imputation process to average out the noise intrinsic in the 10x Visium platform, thereby increasing the absolute values of these correlations. This is consistent with the results presented in Figure 2, where the predicted gene expressions by BLEEP exhibit lower variance compared to the original dataset. This highlights BLEEP's capability to combine expression profiles of similar-looking spots in the dataset and generate expression profiles with reduced noise and enhanced biological signal compared to the original dataset. Ongoing work is evaluating BLEEP's ability to detect new spatially resolved gene modules in response to this observation.

However, an alternate explanation for the observed results could be that averaging during imputation removes genuine, abrupt biological signals, resulting in artificially smoothed expression patterns. It is possible that this may be true for certain genes, especially ones that correlate poorly with image features. Increasing the size of the reference dataset may mitigate this issue by reducing the distance between any query patch and the reference expression profiles. In practice, once the expression encoder is trained, it can be quickly used to integrate new datasets to the shared embedding space as reference. Furthermore, imputed expressions from query datasets could be subsequently integrated, making the process of improving BLEEP prediction natural.

Nevertheless, the improvement offered by BLEEP over existing methods is substantial. We achieved significantly higher correlation with actual expression profiles across the marker gene set (MG), top 50 highly expressed genes (HEG), and top 50 highly variable genes (HVG) (Figure 1), with improvements ranging between 39% to 120% compared to the second highest scoring method. The top predicted genes from BLEEP are consistent with those from ST-Net and HisToGene, but the correlation values (r) are higher by up to 35% compared to the second-highest scoring method. BLEEP also excel in retaining the biological heterogeneity of the original sample as shown in Figures 2, 3, 4 and 5.

Overall, our proposed framework, BLEEP, has the potential to significantly reduce the time and cost associated with gene expression profiling, opening up new avenues for high-throughput analysis of histology images for both research and clinical applications.

## 6 Acknowledgements

All authors thank the Vector Institute, Calcul Québec, and the Digital Research Alliance of Canada for their support. This research has been made possible in part by a grant to G.D.B. from the Chan Zuckerberg Initiative DAF, an advised fund of Silicon Valley Community Foundation. B.W. is supported by the NSERC discovery grant and CIFAR chair programs [RGPIN-2020-06189, DGECR-2020-00294]. R.X. is supported by the Ontario Graduate Scholarship.

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
