# Spatially Resolved Gene Expression Prediction from H&E Histology Images via Bi-modal Contrastive Learning
## Supplemental Material

**Ronald Xie**[1,2,3,4]    **Kuan Pang**[1,2,4]    **Sai W. Chung**[1,5]    **Catia T. Perciani**[1,5]
**Sonya A. MacParland**[1,5]    **Bo Wang**[1,2,3*]    **Gary D. Bader**[1,3,4,6*]

[1]University of Toronto, [2]Vector Institute, [3]University Health Network,
[4]The Donnelly Centre, [5]Toronto General Hospital Research Institute,
[6]Canadian Institute for Advanced Research (CIFAR)
{ronald.xie, kuan.pang, sai.chung,
catia.perciani, gary.bader}@mail.utoronto.ca,
Sonya.MacParland@uhnresearch.ca, bowang@vectorinstitute.ai

## 1 BLEEP Additional Results and Implementation Details

### 1.1 Image Encoder Selection

Table S1 presents the performance evaluation of ResNet and ViT image encoders, along with their corresponding number of parameters. Both ResNet50 and ResNet101 exhibit competitive performance. However, ViT-Base and ViT-Large demonstrate reduced expression prediction accuracy, with fewer genes scoring above 0.3 correlation compared to the original expression profiles. A plausible explanation for this discrepancy is that the utilization of larger models, when combined with a relatively small training dataset ($n = 9269$), may encourage the memorization of information within the network weights rather than effective encoding in the projections. Consequently, the learned joint embedding becomes less effective for downstream imputation in our specific use case.

Supplementary Table 1: The choice of image encoder versus the number of genes with predicted expression correlation $\geq 0.3$ to original.

| Image Encoder | # Parameters | # Genes $\geq 0.3$ corr. |
|---|---|---|
| ResNet50 | 26 M | 20$\pm$1 |
| ResNet101 | 45 M | 20$\pm$2 |
| ViT-Base | 86 M | 7$\pm$2 |
| ViT-Large | 305 M | 2$\pm$2 |

### 1.2 Additional Results

Figure S1 depicts the spatially resolved gene expressions of GLUL and CYP2E1, two key proteins known to be associated with liver zonation [6]. These two genes ranked highly among the top most well predicted genes across all three methods. It can be seen that BLEEP accurately captures the range of variation and the spatial heterogeneity of these genes in clear contrast to HisToGene and ST-Net, where only the mean expression is captured. The variation of the predicted expressions of HisToGene and ST-Net were not well predicted, as evident in the scale range of the color bars in Figure S1. This finding is in accordance with both Figure 2 and Figure 3 from the main text.

---

*Co-senior author

37th Conference on Neural Information Processing Systems (NeurIPS 2023).

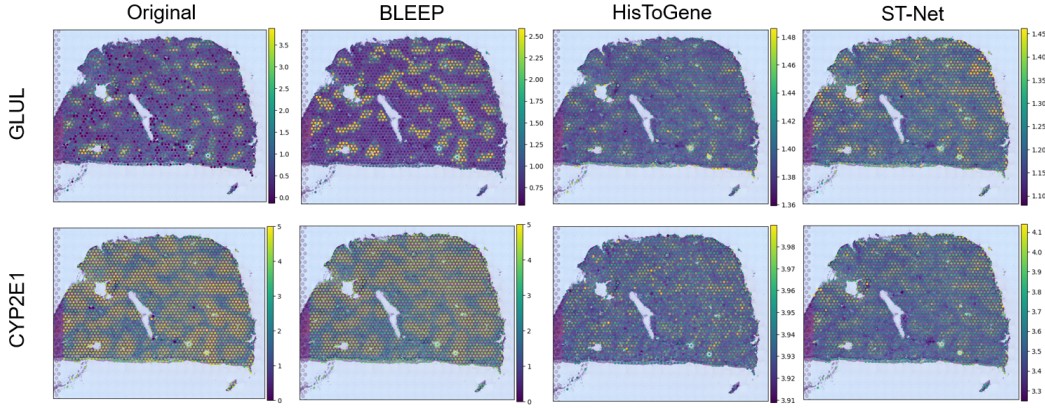

Supplementary Figure 1: Original and predicted spatially resolved expression levels for GLUL (top) and CYP2E1 (bottom) ploted using variable scale and overlaid to the H&E image.

Table S2 presents the results of unsupervised clustering for the predicted expression profiles of spatial spots, compared to the original data. Although HisToGene exhibits higher performance than BLEEP, we caution that it may not be the most appropriate measure for assessing prediction quality due to several reasons. Firstly, the absence of a definitive ground truth for comparison is a challenge. The dataset used in our study primarily consists of human liver hepatocytes, which dominates the biological variations present. The expression of many genes is expected to be similar across tissue spots with relatively small variations, contributing to the difficulty of this application. Consequently, defining discrete clusters for these spatial spots becomes somewhat arbitrary compared to benchmarking datasets used in related studies, which often involve more spatially and expressionally distinct cell types [1, 7, 8].

Furthermore, given the continuous gradient of biological variation in our dataset, the exact clustering method and parameters used significantly impact the definition of discrete clusters. Consequently, while BLEEP does not surpass other methods in terms of clustering metrics such as NMI and ARI, the predicted expression by BLEEP still produces sensible unsupervised clusters that roughly correspond to the periportal and percentral regions of the liver tissue, as demonstrated in Figure 5. Further work involving expert annotation of these slices is required for a more robust comparison.

Supplementary Table 2: NMI and ARI of the predicted expression matrix after clustering

| Method | NMI | ARI |
|---|---|---|
| HisToGene | **0.242**±0.008 | **0.317**±0.008 |
| ST-Net | 0.159±0.029 | 0.185±0.039 |
| BLEEP | 0.186±0.010 | 0.202±0.014 |

### 1.3 BLEEP Default Configuration and Experimental Setting

Here we present the default configuration and experimental setting for BLEEP in Table S3.

Supplementary Table 3: Default configuration and experimental setting for BLEEP.

| config | value |
|---|---|
| image encoder | resnet50 |
| embedding dimension | 2048 |
| projection dim. | 256 |
| # projection layers | 1 |
| batch size | 512 |
| topK | 50 |
| imputation method | weighted avg. |
| optimizer | AdamW |
| base learning rate | 1.0e-4 |
| weight decay | 1.0e-5 |
| optimizer momentum | $\beta_1, \beta_2 = 0.9, 0.0.999$ |

## 2 Comparison Method Implementation Details

We evaluate our method against two of the most commonly cited H&E image-to-expression prediction tools, HisToGene [5] and ST-Net [3]. HisToGene is a vision transformer[2] based model, utilizing neighbouring H&E image patches as input and yielding expression profiles as output. ST-Net is a convolutionally backboned model which processes image tiles for the prediction of gene expressions. In our comparison, we adopt the default architecture configurations as reported in their respective original publications (shown in Supplementary Table 4 and Supplementary Table 5). We also align the experimental settings closely with those used in BLEEP (detailed in 3).

Supplementary Table 4: Default model configurations for HisToGene[5].

| config | value |
|---|---|
| embedding dimension | 1024 |
| transformer layer | 8 |
| attention head | 16 |
| MLP ratio | 2.0 |

Supplementary Table 5: Default model configurations for ST-Net[3].

| config | value |
|---|---|
| convolution backbone | Densenet121[4] |
| embedding dimension | 2048 |