# OpenReview forum: "Spatially Resolved Gene Expression Prediction from Histology Images via Bi-modal Contrastive Learning"
_NeurIPS.cc/2023/Conference — NeurIPS 2023 poster_

### Official Review · Reviewer_J9t4 · 2023-06-21

**Soundness:** 3 good
**Presentation:** 3 good
**Contribution:** 3 good
**Rating:** 6
**Confidence:** 4

**Summary:**

The paper proposes learning a contrastive learning based joint embedding space to align gene expressions and histology. This expression space is used to generate expression predictions for queried patches from the histology modality. The paper shows improved correlations in predicting gene expressions as well as better measure of expression heterogeneity when compared to HisToGene and ST-Net, two popular supervised regression-based methods for gene expression prediction.

**Strengths:**

1. The paper propose a new approach for the problem of predicting gene expression through histology. The problem space is important since sequencing is still expensive, not always available, and its relationship with histology is ripe ground for research in multimodal analysis.
2. The paper seems to be the first one to predict gene expression using bi-modal alignment (although CCA based alignment has been done previously). The reasoning for choosing an alignment-based method is well motivated. The contrastive learning method is simple and effective, and probably overfits less than a direct regression-based method.
3. The correlation results show significantly improved performance on the task of gene expression prediction. Moreover, the results in Figure 2 highlight the very important characteristic of the method not losing out on the heterogeneity information of the genes.


[1] Ash, Jordan T., et al. "Joint analysis of expression levels and histological images identifies genes associated with tissue morphology." Nature communications 12.1 (2021): 1609.

**Weaknesses:**

1. The results show correlations of the reference and predicted gene expressions. However, there's no metrics to show how well the method does for spatially resolving these predictions. The results are also shown on a single dataset.
2. The contrastive learning framework for multimodal problems isn't novel in itself. The smooth loss too has been formulated in different ways in previous work [1, 2, 3]. Thus the main contribution of the paper is its formulation and application to the task of gene expression prediction. The paper is predominantly an application paper, therefore more extensive ablations or more datasets would have been more convincing. The results shown however do show improvements over previous baselines.
3. The paper doesn't motivate choices like smooth contrastive loss, k-nearest neighbor based selection, linear combination to get the imputed genes etc. through empirical ablation experiments.
4. The paper shows the differences between using a fixed scale vs variable scale for spatially resolved predictions. It's useful to motivate why the absolute values for these predictions matter and looking at fixed scale is the way to go.
5. The gene-gene correlation is interesting because HisToGene seems to be closer to the original expression data than the proposed method. The authors note that the method appears to accentuate certain positive and negative correlations, but it's not clear if this is desirable. Similar arguments about the clustering experiment can be made as well where HisToGene seems to do better (although I agree that this isn't a great measure of prediction quality).

[1] Denize, Julien, et al. "Similarity contrastive estimation for self-supervised soft contrastive learning." Proceedings of the IEEE/CVF Winter Conference on Applications of Computer Vision. 2023.
[2] Zheng, Mingkai, et al. "Ressl: Relational self-supervised learning with weak augmentation." Advances in Neural Information Processing Systems 34 (2021): 2543-2555.
[3] Wei, Chen, et al. "Co2: Consistent contrast for unsupervised visual representation learning." arXiv preprint arXiv:2010.02217 (2020).

**Questions:**

1. Which correlation measure is used? Please specify.
2. It's not clear how the plot in Figure 2 was obtained. Is it the mean and variance for each gene across all slides in the test dataset? Why are there multiple points for the predicted expression profile? Some details on this section would be helpful for readers.
3. Can you please share your thoughts on points 1, 3, 4, 5 in the previous section.



**Limitations:**

I believe the authors have shared some limitations in various sections including some seemingly negative results (which is applaudable!). Would really appreciate a commentary on when an alignment based method is expected to do better than a direct regression method and vice versa and explicitly mention some limitations and potential future directions of exploration in this area.

---

> ### Author Rebuttal · Authors · 2023-08-08
>
> We sincerely thank the reviewer for these comments. We summarize the major points below along with our rebuttal to each point.
>
> Correlation Measure:
> - We used the Pearson correlation coefficient for measuring both prediction-GT correlation and gene-gene correlation.
>
> Clarification on Figure 2:
> - Each point on the plot refers to a single gene, whose value reflects its normalized mean or variance across a test sample slice. Blue dots correspond to the ground truth values, while the orange dots represent the inferred values. The gene rankings are performed based on ground truth values in increasing order.
>
> Metrics for spatially resolving the predictions:
> - We provide the qualitative outcomes in Figure 3 and Supplementary Figure 1 to demonstrate BLEEP's proficiency in spatially resolving these predictions. BLEEP generates meaningful unsupervised clusters that align with the periportal and pericentral regions of the sampled liver tissue.
> - For a quantitative evaluation of BLEEP's clustering, we reference the Normalized Mutual Information (NMI) and Adjusted Rand Index (ARI) metrics in Supplementary Table 2. However, due to the continuous gradient of biological variation, the selection of the exact clustering method and its parameters significantly influences the definition of discrete clusters. Therefore, involvement of domain experts to annotate ground truth regions on the H&E image is necessary for a more comprehensive and robust assessment of spatial resolution and planned for future work
>
> Motivation and empirical experiment for using smooth contrastive loss, knn, linear combination for imputation:
> - Please see rebuttal addressed to all reviewers for ablation studies and discussion
>
> Fixed scale vs. variable scale:
> - The importance of fixed scale vs variable scale could be split into two components. Fundamentally, Visium expression data measure the discrete number of transcripts captured that map to each gene. The absolute values in the ground truths provided are therefore meaningful and should be reproduced by any predictive model in terms of both mean and variance. Practically, one may argue that relative expression irrespective of variance is sufficient information for downstream use cases particularly when the counts measured via Visium are themselves subject to a stochastic sampling process during the experiment. However, predicting absolute expression allows meaningful comparisons in magnitude across different genes, which contain richer information than z scores (as each gene has different mean and variance). Furthermore, a fixed scale enables direct comparisons of expression levels across different H&E slides. This approach gives an absolute frame of reference. Visualizing predictions in fixed scale benchmarks the methods’ ability to retain these types of information.
> Lastly, both His2Gene and ST-Net were trained on the same data as BLEEP. The former two approaches’ inability to learn the variance of the held out dataset is a bug not a feature, and reflect shortcomings in the method.
>
> Concern regarding the gene-gene correlation heatmap:
> - The gene-gene correlation heatmap from HisToGene does seem more similar to the original expression. This is likely attributed to BLEEP’s averaging effect during the imputation process. As discussed in the paper, the Visium experimental platform is susceptible to sampling noise. Averaging the expression levels across the top K most similar spots to any query image patch may serve to average out some of this noise and uncover additional biological meaningful signals akin to pseudo-bulking the top K most likely expression profiles for any given query image patch. Supporting this observation, we draw attention to the high and low values in the original gene-gene correlation heatmap. The same correlation patterns are well represented in BLEEP’s heatmap, but just more accentuated, similar to what one would expect from pseudo-bulking. By learning a joint embedding between image and expression, BLEEP is capable of identifying suitable spots to perform pseudo-bulking in a non-biased way and we argue this is desirable for downstream biological inquisition. Nevertheless, such averaging effects may also mask out some genuine biological signals. This limitation is included in the manuscript and requires further investigation.

---

> > ### Comment · Reviewer_J9t4 · 2023-08-16
> >
> > Thanks for the detailed response. I will keep my original rating of a weak accept.

---

### Official Review · Reviewer_WHAP · 2023-07-05

**Soundness:** 2 fair
**Presentation:** 3 good
**Contribution:** 2 fair
**Rating:** 4
**Confidence:** 4

**Summary:**

The authors present, BLEEP, a bi-modal embedding framework capable of generating spatially resolved gene expression profiles of whole-slide H&E stained histology images. This work stems directly from spatial transcriptomics, where we have spatially map gene expression profiles with H&E images. A well descripted tasks is predicting these gene expression profiles from the H&E images alone. This is the task the author set out to address and compare their results with several related methods. Their technical contribution was to apply CLIP-style training plus 'query-reference imputation' to predict expression profiles, rather than relying on disciminative learning alone. They were able to show results that outperform previously reported methods.

**Strengths:**

Strength:
1) The authors introduce the topic well. They do a good job discussing previous work.
2) The paper is well written.
3) The figures are clear and instructive.
4) This is a potentially important biomedical computer vision task.
5) The smoothed target and the query look-up strategy is a novel and intuitive contribution to this task.

**Weaknesses:**

Weaknesses:
1) The paper is, overall, underdeveloped. They have made a small technical contribution and applied this to a very small and narrow domain dataset (normal hepatic tissue). I believe that small technical contributions have an essential place in biomedical AI research, but they must be accompanied by a very strong validation across organ systems, tumor types, platforms, etc. I understand that the availability of these datasets is limited, but the authors have limited their contribution to a single dataset for no clear reason. I would recommend additional experiments in at least two other organ systems, preferably a disease dataset, to demonstrate the generalizability of this method.
2) Minor comment: how are the 'pairs with similar morphological features or expression landscapes' selected? The authors do not specify how these pairs are selected in order to generate the smoothed targets.



**Questions:**

See above.

**Limitations:**

No, there was not a clear description of the limitations or societal impact of the work.

---

> ### Author Rebuttal · Authors · 2023-08-08
>
> We sincerely thank the reviewer for these comments. We summarize the major points below along with our rebuttal to each point.
>
> Additional experiments:
> - While our paper focuses on a single organ system, this choice was made to deeply explore and validate our approach to demonstrate the effectiveness and applicability of the proposed method. There are no technical barriers to extend BLEEP to other organs. Ongoing internal experiments on brain tissue has shown promising results of BLEEPs generalization capacity. We are open to including the results of our experiments in the camera ready version if deemed necessary by the reviewer.  In line with your recommendation, we are currently building foundation models that span multiple organ systems, and it will be released as future work.
>
> Pairs with similar morphological features or expression landscapes:
> - In our approach, we don't explicitly "select" these pairs but rather calculate an internal similarity matrix. This is done for both the morphological features ($H_v$) and the expression landscapes ($H_x$). The internal similarities, $sim(H_v, H_v)$ and $sim(H_x, H_x)$, are computed as the dot product of the feature matrix with its transpose, representing the similarity between every pair of samples. The resultant similarity-adjusted target matrix ($target=(softmax(sim(H_x, H_x) + sim(H_v, H_v) )/2 \cdot \tau)$) allows us to account for the inherent similarities in the data, smoothing the targets based on these similarities.

---

> > ### Author Response · Authors · 2023-08-16
> >
> > To further clarify and provide an update regarding your comment on our limited evaluation, we are actively working to collect more data for testing. However, we need high/full resolution H&E images, as captured by the experimental imaging system (TIFF files as large as 8GB), and we have discovered that only low resolution images are typically shared, forcing us to contact the authors of many studies to access the required high resolution images. Unfortunately, none of the groups we have contacted have been able to provide these high resolution images in a timely manner. This is the main reason we have only included one data set in our manuscript, as this data was available via our collaborators with full resolution images. If we are able to get additional full resolution images in time for NeurIPS deadlines, we will include these as additional tests of our method. Additionally, we believe the current lack of publicly available full resolution images alongside published spatial transcriptomics datasets is a reflection of the field's current lack of means to include these images during data analysis. We hope that our work will raise awareness about the need and value of sharing full resolution images with spatial transcriptomics datasets, and incentivize more researchers to publish these. This will become increasingly important as more spatial transcriptomics analysis methods are developed that also benefit from full resolution image data, and will support the larger community goal of understanding the relationship of transcriptomics and imaging modalities. We will clarify these points in the camera-ready version of the manuscript if successful.

---

### Official Review · Reviewer_QFTu · 2023-07-07

**Soundness:** 3 good
**Presentation:** 4 excellent
**Contribution:** 3 good
**Rating:** 6
**Confidence:** 4

**Summary:**

The authors introduce a method, called BLEEP, of imputing the (aggregate) gene expression profile of cells in patches of histology images. Inspired by CLIP, BLEEP trains image and profile encoders to jointly embed paired images and expression profiles, except in replaces the typical CLIP loss with a novel loss that matches the matrix of similarities between embeddings of image and expression profiles to a matrix of target similarities. Gene expression profiles are imputed for a patch by averaging the expression profiles of reference samples closest in the embedding space. The authors demonstrate SOTA predictive performance over related methods (e.g., HisToGene and ST-Net), while maintaining greater biological and spatial variability, and being less susceptible to batch effects.



**Strengths:**

### Originality

To the best of this reviewer's knowledge, this is the first use of a CLIP-like joint embedding objective for learning to predict gene expression profiles from H&E histology images. Also, the loss introduced by the authors in place of the typical CLIP loss appears to be novel.

### Quality

The work appears to be of sufficient quality to be credible.

### Clarity

The paper is well-written and easy to understand. The discussion is particularly detailed and interesting.

### Significance

Gene expression profile prediction from unstructured biological readouts like H&E images has significant value, since gene expression is interpretable and causal.

**Weaknesses:**

- It is not clear that the loss introduced by the authors, replacing the typical CLIP loss, is required to obtain the reported performance as the the authors claim. A comparison with results obtained using the typical CLIP loss would be useful and should be added.

- The prediction procedure aggregates profiles from a reference dataset, making BLEEP limited in its potential application. It would be interesting to train an image-to-profile decoder and compare the results, especially wrt improved generalization. This is a heavy lift, though, and not required for the revisions.

- As the authors already point out, the aggregation used for query-reference imputation could be removing useful biological signal or have a number of smoothing effects. A discussion of the choices made for imputation and their effects on prediction performance should be added.

- Regarding the previous point, however, there is a lack of details regarding the query-reference imputation step, like number of samples used, method of aggregation.

**Questions:**

- It is not clear if any hyperparameter tuning was done when choosing the default configurations of all models used in this paper. The huge differences in performance make this reviewer wonder if suboptimal parameters were chosen, even if published configurations were used for HisToGene and ST-Net. Please comment.

- It is not clear why HisToGene and ST-Net were chosen for comparison. Please comment.

**Limitations:**

The Discussion section adequately addresses some potential limitations of this work.

---

> ### Author Rebuttal · Authors · 2023-08-08
>
> We sincerely thank the reviewer for these comments. We summarize the major points below along with our rebuttal to each point.
>
> CLIP loss w/ smooth vs. w/o smooth comparison:
> - Please see rebuttal addressed to all reviewers for ablation studies and discussion
>
> Choices made for imputation and their effects on prediction
> - Please see rebuttal addressed to all reviewers for ablation studies and discussion
>
> Details on query-reference steps, implementation details:
> - During query-reference imputation, we agree more details will be useful and plan to include them in the camera-ready version of our manuscript if we are successful during rebuttal. We will expand the discussion on implementation details in the revised version. In BLEEP’s default setting, we use the top 50 most similar samples for query reference imputation, and take their average expression for aggregation.
>
> Settings for the comparison experiments:
> - For HisToGene and ST-Net, we adopted the published configurations, as these parameters had been determined optimal in the original studies. Further, we conducted 3-trial experiments with different random initiations for more robust results. For BLEEP, extensive hyperparameter tuning was done during the development process, e.g. ViT backbone vs. Conv. Backbone for image encoder. Therefore, the performance differences are not due to manually chosen suboptimal parameters but reflect the inherent differences in these approaches. We will include evaluation scripts in the code release for community replication.
>
> Comparison with HisToGene and ST-Net:
> - HisToGene and ST-Net are state of the art methods for the task of expression prediction from images in this field and are not weak comparisons. The performances of these methods highlight the difficulty of the image to expression task and are competitive reflections of the current state of the field to our best knowledge.
> - We surveyed many other related methods such as HE2RNA[1], hist2RNA[2], HIST2ST[3], and more recently, CeLEry[4] and SCHAF[5]. However they are not 1 to 1 comparisons that either take different information as input or tackle different settings with different use cases. For example, SCHAF requires both H&E image and the paired single cell RNA sequencing data rather than spatial transcriptomics data, the former containing no spatial information but higher expression resolution. In short all these surveyed methods were carefully evaluated and deemed not suitable to include as comparisons.
>
> [1] Schmauch, Benoît, et al. "A deep learning model to predict RNA-Seq expression of tumours from whole slide images." Nature communications 11.1 (2020): 3877.
>
> [2] Mondol, Raktim Kumar, et al. "hist2RNA: An efficient deep learning architecture to predict gene expression from breast cancer histopathology images." Cancers 15.9 (2023): 2569.
>
> [3] Zeng, Yuansong, et al. "Spatial transcriptomics prediction from histology jointly through transformer and graph neural networks." Briefings in Bioinformatics 23.5 (2022): bbac297.
>
> [4] Zhang, Qihuang, et al. "Leveraging spatial transcriptomics data to recover cell locations in single-cell RNA-seq with CeLEry." Nature Communications 14.1 (2023): 4050.
>
> [5] Comiter, Charles, et al. "Inference of single cell profiles from histology stains with the Single-Cell omics from Histology Analysis Framework (SCHAF)." BioRxiv (2023): 2023-03.

---

> > ### Comment · Reviewer_QFTu · 2023-08-21
> >
> > I would like to thank the authors for their rebuttal and attempts at clarifying the choices made. I will maintain my current rating.

---

### Official Review · Reviewer_VLr1 · 2023-07-12

**Soundness:** 2 fair
**Presentation:** 4 excellent
**Contribution:** 3 good
**Rating:** 5
**Confidence:** 3

**Summary:**

The authors have developed the model titled BLEEP, which is a contrastive learning implementation, trained on data from the 10x Genomics Visium platform, a common spacial genomics platform.  Spacial genomics generates high dimensional data that includes both images and RNA expression on small patches on tissue slides.   BLEEP uses two destinct encoder for the image and expression data.  These embeddings are then used as inputs for contrastive learning by a modified CLIP method. The approach of contrastive learning is sound given the high feature status in both the data which is being learned from and the images that are to be predicted.

The model is tested vs. benchmark models on marker gene, highly variable genes, and highly expressed genes and model shows modest increase in correlation to other models.  BLEEP is also compared to how well gene expression is predicted for several genes. Figure 2 illustrates that the BLEEP model does show a better correlation to gene expression variation than other models. Additional figures show the heatmaps of actual relative to model predictions for fixed and variable scaled outputs.  Section 4.3 discusses how the model is perhaps more robust to artifactual portions of images.





**Strengths:**

I appreciate the color expressed around how "ill-posed" the problem is that the authors are trying to address.  Likewise, there is a huge dimensionality challenge and the sequencing methods are rather flimsy as stated.  Going into this problem realizing the challenge is very good.

The contrastive learning approach is likely the best approach the the task at hand.

The presentation of the data is not to express absolute confidence in the results and the authors only suggest that perhaps the methods are more sound given that the task itself may not be a solvable task.

**Weaknesses:**

There is no guarantee that the signal that is being derived from given methods is in learning from the data presented. While the presentation of improved variance prediction is probably the strongest evidence in this regard.


Even though the model appears to outperform benchmarks, the correlations remain very low and likely the predictions are not particularly useful  task.

**Questions:**

Can you foresee any circumstance where the model could be used on image data alone to provide information at a low cost relative to running the spacial genomics platform.



**Limitations:**

The main limitation is that this may not be a useful task to pursue.  However, it does seem authors are aware of this limitation and if we don't try to tackle what seems like an impossible task then it is hard to make progress.

---

> ### Author Rebuttal · Authors · 2023-08-08
>
> We sincerely thank the reviewer for these comments. We summarize the major points below along with our rebuttal to each point.
>
> There is no guarantee that the signal that is being derived from given methods is in learning from the data presented. While the presentation of improved variance prediction is probably the strongest evidence in this regard:
> - Our proposed method aims to learn a joint representation between H&E image and expression from the data presented. The improved variance prediction relative to other methods may be attributed to a well-learned joint embedding in conjunction with our choice of the query-reference inference strategy
>
> Even though the model appears to outperform benchmarks, the correlations remain very low and likely the predictions are not particularly useful task:
> - We thank the reviewer for this important question. We are aware that full expression prediction from image features is likely ill-posed. However we do believe there is substantial mutual information between image features and a subset of gene expressions. Our proposed contrastive learning objective helps prioritize these genes without any injection of prior knowledge. As seen in Table 2. The most well predicted genes for this dataset are also functional and well documented by prior biological research. For example, CYP1A2 is an oxidizing enzyme that is also widely cited as a liver zonation marker. Furthermore, the top predictions from BLEEP are rather consistent with top predictions by His2Gene and ST-Net, but with higher correlations to original expression, highlighting a step in a promising direction.
>
> Can you foresee any circumstance where the model could be used on image data alone to provide information at a low cost relative to running the spatial genomics platform:
> - Yes we believe our work may stimulate others to iterate upon our efforts and ultimately make advances towards this goal.
> - As it stands currently, we chose to benchmark our method on 3 sets of genes (Marker genes, highly expressed genes and highly variable genes). We believe these 3 sets of genes are likely more enriched for genes useful for diagnosis and treatment (drug targets). Within these categories while average correlation hovers between 0.17-0.21, some subset of these genes are quite reasonably predicted and potentially useful for aforementioned applications. However we agree more work is needed to further examine the potential of BLEEP in this use case. We are undergoing followup experiments to examine the effect of increased reference size and working on expanding the reference to cover multiple tissue types
> - Furthermore, this method could pave the way for the construction of more comprehensive foundation models for spatial transcriptomics analysis [1]
>
> [1] Cui, Haotian, et al. "scGPT: Towards Building a Foundation Model for Single-Cell Multi-omics Using Generative AI." bioRxiv (2023): 2023-04.

---

> > ### Comment · Reviewer_VLr1 · 2023-08-16
> >
> > Thank you for your sound and clarifying comments. I maintain my official review scores.

---

### Official Review · Reviewer_XnkF · 2023-07-23

**Soundness:** 2 fair
**Presentation:** 3 good
**Contribution:** 3 good
**Rating:** 5
**Confidence:** 3

**Summary:**

This paper studies the problem of gene expression profiling using histology images. They propose a bi-modal embedding framework BLEEP (Bi-modaL Embedding for Expression Prediction), which is capable of generating spatially resolved gene expression profiles of whole-slide Hematoxylin and eosin (H&E) stained histology images. The proposed method can significantly reduce the time and cost associated with gene expression profiling.

**Strengths:**

1. The paper utilizes a deep learning method to tackle the problem of gene expression prediction, reducing the time and cost.
2. The paper is well-written and easy to follow. The motivation is clearly illustrated.
3. The experiment design looks reasonable and interesting. Supplementary materials illustrate details of experimental settings.

**Weaknesses:**

1. Weak comparisons. This work only compares with two deep learning methods HisToGene and ST-Net. There should be other deep learning methods about image processing that can be applied to this area and the authors should compare with more baselines to confirm the efficacy of the proposed method.
2. Limited impact of the proposed method. Although the problem in this paper is interesting, this work should discuss broad impact of the proposed method. For instance, how can we apply this method to other biomedical applications.

**Questions:**

See weaknesses.

**Limitations:**

The proposed method can effectively address the problem of gene expression prediction. However, it is not clear whether we can apply this method to more applications.

---

> ### Author Rebuttal · Authors · 2023-08-08
>
> We sincerely thank the reviewer for these comments. We summarize the major points below along with our rebuttal to each point.
>
> Weak comparisons. This work only compares with two deep learning methods HisToGene and ST-Net:
> - HisToGene and ST-Net are state of the art methods for the task of expression prediction from images in this field and are not weak comparisons. The performances of these methods highlight the difficulty of the image to expression task and are competitive reflections of the current state of the field to our best knowledge.
> - We surveyed many other related methods such as HE2RNA[1], hist2RNA[2], HIST2ST[3], and more recently, CeLEry[4] and SCHAF[5]. However they are not 1 to 1 comparisons that either take different information as input or tackle different settings with different use cases. For example, SCHAF requires both H&E image and the paired single cell RNA sequencing data rather than spatial transcriptomics data, the former containing no spatial information but higher expression resolution. In short all these surveyed methods were carefully evaluated and deemed not suitable to include as comparisons.
>
> [1] Schmauch, Benoît, et al. "A deep learning model to predict RNA-Seq expression of tumours from whole slide images." Nature communications 11.1 (2020): 3877.
>
> [2] Mondol, Raktim Kumar, et al. "hist2RNA: An efficient deep learning architecture to predict gene expression from breast cancer histopathology images." Cancers 15.9 (2023): 2569.
>
> [3] Zeng, Yuansong, et al. "Spatial transcriptomics prediction from histology jointly through transformer and graph neural networks." Briefings in Bioinformatics 23.5 (2022): bbac297.
>
> [4] Zhang, Qihuang, et al. "Leveraging spatial transcriptomics data to recover cell locations in single-cell RNA-seq with CeLEry." Nature Communications 14.1 (2023): 4050.
>
> [5] Comiter, Charles, et al. "Inference of single cell profiles from histology stains with the Single-Cell omics from Histology Analysis Framework (SCHAF)." BioRxiv (2023): 2023-03.
>
> This work should discuss broader impact of the proposed method. For instance, how can we apply this method to other biomedical applications:
> - We currently have no plans of applying this method to other biomedical applications. However, we are dedicating our efforts to extend this work to cover multiple organs and modalities, which we anticipate will help further improve performance and generalizability of BLEEP for prediction expression based on H&E image.
> - In terms of broader impact, we anticipate that BLEEP may improve biological understanding of h&e images by drawing connections between image features and gene expression. H&E staining is a ubiquitously used experimental technique in biology, our work may allow further understanding of H&E stained samples through learning the joint embedding.  Furthermore, clinical classification in the field of pathology could benefit from better understanding of image features in terms of molecular features, which underlie disease. Lastly, understanding the relationship of images and gene expression will help projects like the human cell atlas [6] to create coordinate frameworks that position genes spatially in terms of tissue anatomy, and ultimately in terms of the whole body.
>
> [6] Rozenblatt-Rosen, Orit, et al. "The Human Cell Atlas: from vision to reality." Nature 550.7677 (2017): 451-453.

---

> ### Comment · Reviewer_XnkF · 2023-08-16
>
> Thanks for the rebuttal. I think the authors have addressed my concerns as they have explained their broader impact in healthcare. I would like to raise my score.

---

### Author Rebuttal · Authors · 2023-08-08

We sincerely thank the reviewers for their comments and their acknowledgment to the strengths of our paper including:
- “significantly improved performance” and “not losing out on the heterogeneity information.” (J9t4).
- “first use of a CLIP-like joint embedding objective for learning to predict gene expression profiles from H&E histology images” (QFTu)
- “The smoothed target and the query look-up strategy is a novel and intuitive contribution” (WHAP)
- “Contrastive learning approach is likely the best approach to the task at hand” (VLr1)
- “Experiment looks reasonable and interesting” (XnkF)

We have performed the requested ablation experiments from some of the reviewers. We share the resulting ablation table in the attached PDF below. From this experiment we make a few important observations:

- The choice of K during the query-reference imputation process influences the prediction quality quite negatively when a low value is selected (K = 10). Values of K above our default value could provide some small improvements to correlation of the resulting predicted expression values for the HVG and HEG gene sets, but the differences were not pronounced. This is interesting but inline with what one might expect from taking the pseudo-bulk of the top K most likely expression profiles given any query image. However, the MG gene set did not show much improvement with increasing K. Furthermore, doing so may carry a trade off of further systematically deviating from the original variance of the dataset due to the increased averaging effect. With this in mind, we feel our default value of K = 50 remains adequate.

- The most similar match between query and reference is usually not the best prediction (as seen from the 3rd row of the ablation table and indirectly the 4th row when predictions are weighted by their similarity). However we suspect the gap may close to some degree as the reference grows further in size, but in general some amount of averaging is desired for query-reference imputation to remove some noise intrinsic in the Visium platform. However, in the discussion section of the manuscript, we further highlight the possibility of genuine biological signals being averaged out, which is an important consideration to be further investigated.

- Smoothing the contrastive loss objective to take into account patch similarity showed modest increase in performance. The gain in performance may be due to the fact that relaxing the contrastive objective is more compatible with the similarity based inference strategy. The smoothing may help lessen the extent similar references are pushed apart in embedding space during training, resulting in improved querying of the top K most likely expression profiles given a query image patch during the inference stage.

Lastly, for all reviewers, we want to further motivate BLEEP and clarify our vision regarding future extensions of BLEEP and the project’s potential impact:

- The task of spatially resolved, transcriptome-wide expression prediction given an H&E stained image is likely extremely difficult. BLEEP offers an unbiased way to prioritize genes that are more likely to be well predicted from image features via the learning of a joint embedding between image and expression features. It also tackles the curse of dimensionality through the use of query-reference imputation while simultaneously alleviating some technical noise intrinsic to the Visium platform. These design choices allowed us to see significant improvements over state-of-the-art methods such as ST-Net and His2Gene (upwards of 120% increase in correlation with ground truth expression).
- BLEEP is currently restricted for research purposes only and still has a lot of room for improvement. Nevertheless, BLEEP may already be immediately useful for gaining biological understanding of H&E images due to the connections drawn between image and expression features. H&E staining is a ubiquitously used experimental technique in biology, our work may allow further understanding of H&E stained samples through learning the joint embedding.  Similarly, clinical classification in the field of pathology could benefit from better understanding of image features in terms of molecular features, which underlie disease. Lastly, understanding the relationship of images and gene expression will help projects like the human cell atlas [1] to create coordinate frameworks that position genes spatially in terms of tissue anatomy, and ultimately in terms of the whole body.
- While our paper focuses on a single organ system, this choice was made to deeply explore and validate our approach to demonstrate the effectiveness and applicability of the proposed method. There are no technical barriers to extend BLEEP to other organs. Ongoing internal experiments on brain tissue have shown promising results of BLEEP’s generalization capacity. We are open to including the results of our experiments in the camera ready version if deemed necessary by the reviewer. In line with the recommendation of some reviewers, we have already begun efforts towards building foundation models that span multiple organ systems, and it will be released as future work.
- Finally, we believe expert annotation of the different regions of the H&E image will provide us with better ground truths for tissue-wide expression pattern benchmarking. This will be planned for future work.

[1] Rozenblatt-Rosen, Orit, et al. "The Human Cell Atlas: from vision to reality." Nature 550.7677 (2017): 451-453.

---

### Decision · Program_Chairs · 2023-09-21

**Decision:**

Accept (poster)

**Comment:**

The paper addresses an interesting problem of predicting gene expression profiling from histological images, with significant impact to various biomedical applications. The initial review ratings were mixed. The reviewers raised concerns on the interpretation of results (low correlation with transcriptomics data), and the experimental evaluation (results on a single dataset with one tissue type). They also questioned many technical details, and pointed out potential issues in ethics. The author provided a detailed response. Most of the concerns have been addressed, resulting in favorable final ratings from the reviewers. One remaining concern lies in the consideration of a single dataset, which the authors also admitted yet explained the difficulty of acquiring spatial transcriptomics with high resolution imaging data.

The AC considers that the paper introduced a challenging biomedical problem, presented a promising solution, and is thus worth accepting to NeurIPS. The AC encourages the authors to further address the review comments and strengthen the paper.